# Oropharyngeal microbiome profiling and its association with age and heart failure in the elderly population from the northernmost province of China

Jian Liu,[1] Xiao-Yu He,[1] Ke-Laier Yang,[2] Yue Zhao,[3] En-Yu Dai,[1] Wen-Jia Chen,[3] Aditya Kumar Raj,[1] Di Li,[1,4,5] Min Zhuang,[1,4,5] Xin-Hua Yin,[3,6] Hong Ling[1,4,5]

**ABSTRACT**    Respiratory tract infections are the most common triggers for heart failure in elderly people. The healthy respiratory commensal microbiota can prevent invasion by infectious pathogens and decrease the risk of respiratory tract infections. However, upper respiratory tract (URT) microbiome in the elderly is not well understood. To comprehend the profiles of URT microbiota in the elderly, and the link between the microbiome and heart failure, we investigated the oropharyngeal (OP) microbiome of these populations in Heilongjiang Province, located in the North-East of China, a high-latitude and cold area with a high prevalence of respiratory tract infection and heart failure. Taxonomy-based analysis showed that six dominant phyla were represented in the OP microbial profiles. Compared with young adults, the OP in the elderly exhibited a significantly different microbial community, mainly characterized by highly prevalent *Streptococcus*, *unidentified_Saccharibacteria*, *Veillonella*, *unidentified_Pre votellaceae*, and *Neisseria*. While *unidentified_Prevotellaceae* dominated in the young OP microbiome. There was competition for niche dominance between *Streptococcus* and member of Prevotellaceae in the OP. Correlation analysis revealed that the abundance of *unidentified_Saccharibacteria* was positive, while *Streptococcus* was negatively correlated to age among healthy elderly. The bacterial structure and abundance in the elderly with heart failure were much like healthy controls. Certain changes in microbial diversity indicated the potential OP microbial disorder in heart failure patients. These results presented here identify the respiratory tract core microbiota in high latitude and cold regions, and reveal the robustness of OP microbiome in the aged, supplying the basis for microbiome-targeted interventions.

**IMPORTANCE** To date, we still lack available data on the oropharyngeal (OP) microbial communities in healthy populations, especially the elderly, in high latitude and cold regions. A better understanding of the significantly changed respiratory tract microbiota in aging can provide greater insight into characteristics of longevity and age-related diseases. In addition, determining the relationship between heart failure and OP microbiome may provide novel prevention and therapeutic strategies. Here, we compared OP microbiome in different age groups and elderly people with or without heart failure in northeastern China. We found that OP microbial communities are strongly linked to healthy aging. And the disease status of heart failure was not a powerful factor affecting OP microbiome. The findings may provide basic data to reveal respiratory bacterial signatures of individuals in a cold geographic region.

**KEYWORDS**    healthy aging, respiratory tract, heart failure, oropharyngeal microbiome

**Ad Hoc Peer Reviewers** Baohong Wang, The First Affiliated Hospital of Medicine School, Zhejiang University, Hangzhou, China; Lili Ren, Institute of Pathogen Biology, Chinese Academy of Medical Sciences & Peking Union Medical College, Beijing, China

Address correspondence to Hong Ling, lingh@ems.hrbmu.edu.cn, or Xin-Hua Yin, xinhua_yin@163.com.

Jian Liu, Xiao-Yu He, Ke-Laier Yang, and Yue Zhao contributed equally to this article. Author order was determined in order of increasing seniority.

The authors declare no conflict of interest.

See the funding table on p. 16.

The respiratory tract is a complicated organ system that can be divided into upper (URT) and lower (LRT) respiratory tract. The URT includes the anterior nares, nasal cavity, nasopharynx (NP), and oropharynx (OP) above the vocal cords. The colonization and overgrowth of pathogens in the URT are the initial stages of causing upper, lower, or disseminated respiratory tract infections (1). Inhibiting this first step of pathogenesis for respiratory infections by the resident microbiota, also known as "colonization resistance," may play a crucial role in maintaining respiratory health.

The OP is a crucial link between the LRT, gastrointestinal tract, and other parts of the URT, making it susceptible to a wide variety of external and internal microorganisms and characterized by a high prevalence of *Streptococcus*, *Neisseria*, as well as anaerobic bacteria such as *Veillonella* and *Prevotella* (2, 3). The microbial migration to the lung primarily occurs through micro-aspiration in the URT (4). High-throughput studies have confirmed that the lung microbiome in healthy individuals closely resembles the URT (5–7). Among them, it appears that the OP ecological niche serves as the primary source of the lung microbiome (5). Although the lung microbiota could be a combination of both OP and NP bacterial communities, the differences between OP and lung microbiota were found to be smaller than those between NP and lung (7, 8). These findings reflect a close relationship between OP and LRT microbiota. Currently, the OP microbiota, with its unique distribution characteristics, has become a focal point in microbiome studies.

In recent years, there has been an accumulation of evidence linking OP bacterial communities to pneumonia (3, 9), respiratory virus infection including severe acute respiratory syndrome coronavirus 2 (SARS-CoV-2) (10–15), and other respiratory diseases (16–18). The OP microbiota in patients was identified as containing a lower microbial diversity and disordered microbial community structure (19). The OP microbiome could potentially serve as a predictor for disease susceptibility. Tsang et al. discovered that certain members of the genera *Streptococcus* and *Prevotella* were associated with susceptibility to influenza virus infection (20). Particularly, several researchers have successfully identified OP indicator microorganisms for SARS-CoV-2, which could potentially be used as biomarkers to distinguish and regulate the severity of disease (13, 15, 21). If these associations are causal, then regulating the microbiome could potentially reduce the risk of related diseases. Therefore, analyzing the OP microbiome will contribute to our understanding of how respiratory diseases develop and progress.

The elderly who undergo immune senescence are the most susceptible to respiratory infectious diseases. For instance, age-associated chronic inflammation augments the susceptibility and severity of pneumonia in the elderly (22). However, the OP microbiome of the elderly, including long-lived individuals, is largely undocumented. A case–control study has shown that the genera *Streptococcus* and *Veillonella* were significantly more prevalent in adults who passed the age of 65 years (23). These findings demonstrated that specific microbial profiles of the OP were associated with higher age. The OP microbiome in older patients with lower limb fracture (24), chronic obstructive pulmonary disease (25), and pneumonia (3) has its own remarkable characteristics. Moreover, in the elderly, the microbiota within the OP more closely resembles that of the anterior nares (26). These results possibly explain the loss of topography in microbial communities in the elderly. Hence, a better understanding of significant variations in the aging microbiome may provide insight into longevity and age-related diseases.

Cardiovascular diseases, including heart failure, are a major cause of morbidity and mortality in the elderly. According to a limited number of epidemiological reports, the prevalence of heart failure in Asian populations ranged from 1.26% to 6.7% (27, 28). And chronic heart failure was more prevalent among the elderly in China (29). The province of Heilongjiang is located in the northeast of China and is the coldest area in the country. The prevalence of heart failure in the northern areas of China was significantly higher than in the south (30). A survey of 1,862 chronic heart failure, admitted to 20 tertiary hospitals in Heilongjiang Province, identified an in-hospital mortality rate of 4.7% (31). Further, respiratory tract infections served as the most common and important triggers for heart failure, aggravating symptoms (30). Therefore, an exploration of the makeup of

the OP microbiota in the elderly with heart failure could be vital to our understanding of the effect of respiratory infection on heart failure disease.

Our knowledge of the colonization and succession of URT bacterial communities in healthy populations in high latitude and cold regions is lacking. And no reports have focused on the OP microbiota from patients with heart failure. In particular, it is very important and urgent to understand the microecological characteristics of the elderly who are prone to respiratory tract infection and thus heart failure to maintain respiratory tract health. In this study, we used 16S rRNA gene amplicon sequencing, a molecular profiling approach, to produce a more complete picture of microbial colonization. We investigated the OP microbiome of healthy young and elderly individuals in the coldest region of China to show shifts in bacterial communities that occur with age. We also described the OP bacterial communities in the context of heart failure. The result provides a comprehensive look at targets for prevention and treatment when the respiratory tract microbiome is dysregulated.

## MATERIALS AND METHODS

### Study design

To compare the OP microbiome profile of healthy people at different ages, 42 elderly individuals aged ≥65 years who provided oropharyngeal samples during the medical examination at the Medical Center of Harbin Medical University and 30 college students from Harbin Medical University were recruited. To analyze the OP microbiome profile at condition of heart failure, the patients with heart failure aged ≥65 years, who were admitted to the First Affiliated Hospital of Harbin Medical University, were invited to participate in the study. For all the patients, a hospitalized diagnosis of heart failure was given by a cardiologist and grades III and IV of heart failure were assessed according to the criteria of the New York Heart Association (NYHA).

Healthy elderly people were eligible if: (i) no use of antibiotics, antifungal drugs, antiviral drugs, antiparasitic drugs, corticosteroids, cytokines, antineoplastic drugs, immunosuppressive drugs, and probiotics in the past 3 months; (ii) age ≥65 years old; and (iii) never smoking. Exclusion criteria for healthy elderly people were as follows: (i) oropharyngeal deformities or tumors, underlying diseases such as asthma; pharyngitis, rhinitis, sinusitis, and so on; (ii) transient syncope in the past 3 months; (iii) symptoms that affect breathing such as heart failure; (iv) influenza vaccination in the last 3 months; (v) previous history of infectious diseases such as tuberculosis, hepatitis B, and hepatitis C; and (vi) acute illness with or without fever in the past 3 months. In addition, all subjects were natives of Heilongjiang Province and were involved in the study between November and April. All participants signed informed consent and demographic information was gathered. This study was approved by the Ethics Committee of Harbin Medical University (Harbin, China) (HMUIRB20180002).

### Sampling and DNA isolation

OP samples were collected using Copan-flocked swabs (Copan, Italy) according to respiratory specimen collection guidelines from the Centers for Disease Control and Prevention (CDC). The swab was opened and vibrated for several seconds in the sampling chamber as a negative control to evaluate potential contamination. All swabs were immediately placed in 400 µL Tris-ethylenediaminetetraacetic acid (EDTA; Biotopped, China), in which 20.8 µL lysozyme was added to a final concentration of 5 mg/mL. The mixture was then vortexed at 37℃ for 30 min, followed by the extraction of genomic DNA using the protocol from QIAamp DNA Mini Kit (QIAamp, Germany).

### Amplicon library preparation and Illumina sequencing

The forward primer 336F (5′-CGATGTGTACTCCTACGGGAGGCAGCA-3′) and the reverse primer 806R (5′-GTGGACTACHVGGGTWTCTAATACTGAT-3′) were used to amplify the

16S rRNA gene V3–V4 hypervariable region to construct a PCR-amplified library. The amplification mix contained 1*PrimeSTAR HS (Premix) (Takara, China) and 0.25 mM of each primer. The PCR procedure included: initial denaturation (94°C; 5 min), followed by 35 cycles of denaturation (94°C; 1 min), annealing (57°C; 25 s), and extension (72°C; 30 s), final extension (72°C; 10 min). The PCR products were electrophoresed in a 2% agarose gel and those with bands between 400 and 450 bp were chosen for further purification with the GeneJET Gel Extraction Kit (Thermo Scientific, USA).

Sequencing libraries were generated using the TruSeq DNA PCR-Free Sample Preparation Kit (illumine, USA) with index codes added as recommended by the manufacturer. The library quality was assessed with a Qubit@2.0 Fluorometer and Agilent Bioanalyzer 2100 system. Amplicon pyrosequencing was performed with an Illumina HiSeq 2500 to produce 250 bp paired-end reads.

## Data processing

Paired-end reads of the original DNA fragments were merged using FLASH (32), a fast and accurate analysis tool designed to incorporate paired-end reads when the original DNA fragments were shorter than twice the length of the reads. Sequencing reads were assigned to each sample on the basis of unique barcodes. Quantitative insights into microbial ecology (QIIME, version 1.9.1) was used to process and analyze the high-quality sequences (33).

In brief, reads were initially filtered by QIIME quality filters to obtain the high-quality clean tags. Then, tags were compared with the Gold database to detect chimera sequences using the UCHIME algorithm (34) to detect chimera sequences. Finally, the chimera sequences were removed and the effective tags were obtained. Operational taxonomic unit (OTU) screening was then performed using a *de novo* OTU picking protocol with a 97% similarity threshold. A representative sequence for each OTU was picked with the Remote Desktop Protocol (RDP) classifier (35), which was used to assign taxonomic data for each representative sequence, and the annotation was based on the SILVA database (36). Bacterial diversity was determined using sampling-based analysis of OTUs, and was displayed as a rarefaction curve. Bacterial richness and diversity across the samples were calculated using the following indices: Chao 1; abundance-based coverage estimators (ACE); Simpson; and Shannon. Principal coordinate analysis (PCoA) using unweighted and weighted UniFrac distance matrices or principal component analysis (PCA) using Euclidean metric was used to visualize the differences among bacterial communities of different samples. And these distances were also visualized by the unweighted pair-group method with arithmetic means (UPGMA). UPGMA clustering is a type of hierarchical clustering method using average linkage, and can be used to interpret the distance matrix. Linear discriminant analysis (LDA) coupled with effect size (LEfSe) was performed to identify the bacterial taxa differentially represented between groups. LEfSe was used to identify biomarkers within the OP microbiome across groups at multiple levels in data sets, grade the biomarker according to statistical significance, and visualize the results using taxonomic bar charts and cladograms.

## Statistical analysis

Statistical analyses and graphing were performed using GraphPad Prism software (version 6.0). All table data are presented as mean ± standard error of mean (SEM). The results of OP microbiome diversity indices were analyzed using the non-parametric Kruskal-Wallis *H* test, followed by Dunn's multiple comparisons test for three groups or the Mann-Whitney *U* test for two groups. The relative abundance of a microbe in a sample was calculated by normalizing the read count to the total read in the sample. Differences in relative abundances of phyla and genera among groups for statistical significance were determined using Metastats analysis. Correlations between bacterial abundance and age were assessed by Spearman's correlation analysis. A *P*-value <0.05 was considered statistically significant.

## RESULTS

### Study population

A total of 104 subjects from Heilongjiang province in China were analyzed for the oropharyngeal bacterial community profiles. The characteristics of the study population are summarized in Table 1. These subjects were divided into two teams. One was the age team of healthy individuals, which was divided into three groups: Young (30 young adults, aged 22 and 26 years), Elderly (16 younger elderly, aged 65–74 years), and Older (26 older elderly, aged 75 and 95 years). The cutoff of 75-year-olds was in keeping with the previous research (37–39). Another was the disease team, which was divided into two groups: heart failure (HF, 32 patients, aged 65 and 83 years) and healthy group (H, 32 healthy individuals from age team who were age-matched with the heart failure subjects).

In general, subjects' characteristics, including body mass index (BMI) and gender proportions, were comparable among the groups, except for several risk factors or comorbidities known to be associated with heart failure, such as smoking, diabetes mellitus, and hypertension.

### Quality control of the sequences

After removing noisy reads, chimeras, and singletons, a total of 3,779,374 and 3,025,853 effective reads remained in the age and disease team, respectively. The distribution of reads was between 29,711 and 69,796 per sample. The Good's coverage, which represents the sequence depth of the OP microbiome, was ranged from 0.990 to 0.999 (Table S1). Rarefaction curve of observed species per group showed that the mean number of observed species reached a plateau at ~30,000 sequence reads (Fig. 1A; Fig. S1A), and species accumulation curve of observed species per sample also plateaued (Fig. 1B; Fig. S1B). This indicated that almost all OTUs present in each group were detected, and that about 30,000 reads were sufficient to identify most of the microbial community members within the OP microbiome.

### Characteristics of bacterial taxonomic composition in different age groups

We first evaluated the bacterial taxonomic composition at the phyla level. Six predominant phyla were represented in the OP microbial profiles: Firmicutes, Bacteroidetes, Actinobacteria, Saccharibacteria (formerly known as TM7), Fusobacteria, and Proteobacteria. The remaining phyla were presented at much lower relative abundance (Fig. 2; Table S2). Taxonomic profiles were illustrated using UPGMA hierarchical clustering

**TABLE 1** Characteristics and clinical parameters of two teams[c]

| Characteristics | Age team | | | Disease team | |
| --- | --- | --- | --- | --- | --- |
| | Young (n = 30) | Elderly (n = 16) | Older (n = 26) | H (n = 32) | HF (n = 32)[a] |
| Age, mean (range), years | 23.56 (22–26) | 68.63 (65–74) | 82.54 (75–95) | 73.66 (65–83) | 72.50 (65–83) |
| Male:female | 11:19 | 6:10 | 13:13 | 12:20 | 18:14 |
| BMI, mean (SD), kg/m$^2$ | 20.91 (2.66) | 23.84 (3.67) | 23.44 (3.13) | 23.75 (3.38) | 22.78 (3.65) |
| Smoking (n) | 0 | 0 | 0 | 0 | 13 |
| LVEF, mean (SD), % | NA | NA | NA | NA | 40.72 (11.81) |
| BNP, mean (SD), pg/mL | NA | NA | NA | NA | 925.94 (796.93) |
| Heart failure grade | | | | | |
| Grade III | NA | NA | NA | NA | 6 |
| Grade IV | NA | NA | NA | NA | 26 |
| Comorbidity | | | | | |
| Diabetes mellitus (n) | 0 | 0 | 1 | 0 | 10 |
| Hypertension (n)[b] | 0 | 0 | 5 | 3 | 16 |

[a]All HF patients were diagnosed III and IV stages.
[b]Hypertension: diastolic and systolic blood pressure were ≥18.7 kPa (140 mmHg) and ≥12.0 kPa (90 mmHg), respectively. H, health group; HF, heart failure group.
[c]Body mass index, BMI; brain natriuretic peptide, BNP; left ventricular ejection fraction, LVEF; not applicable, NA.

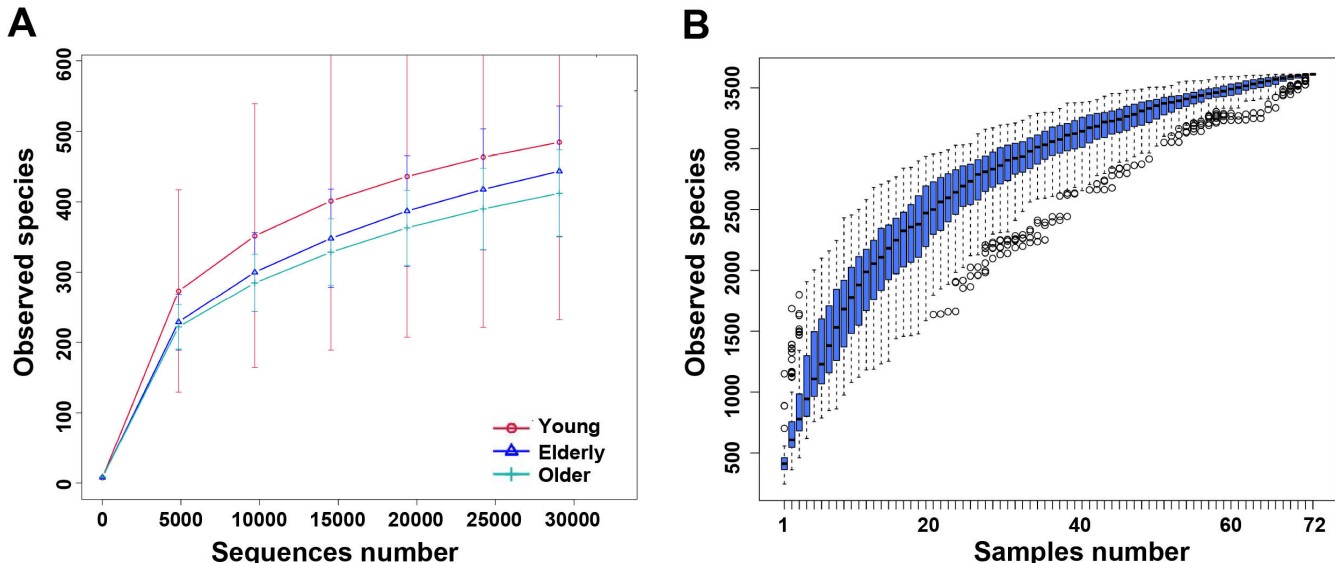

**FIG 1** Rarefaction curve and accumulation curve of observed species of oropharyngeal microbiota among healthy individuals. (A) Rarefaction analysis of bacterial 16S rRNA gene sequences was used to evaluate whether further sequencing would likely detect additional taxa, indicated by a plateau. The *y*-axis denotes the number of operational taxonomic units detected by Hiseq sequencing at the corresponding sequencing depths shown along the *x*-axis. (B) Observed species index curves evaluating the number of samples likely required to identify additional taxa, indicated by a plateau. The *y*-axis denotes the richness detected by Hiseq sequencing at the corresponding number of samples shown along the *x*-axis.

based on weighted UniFrac distance. The resulting tree, shown in Fig. 2, branched into two main clades. The Older group was clustered with the Elderly group, while the Young group was separated from them. The relative abundance of major phyla was ranked differently in three age groups. The OP dominant phyla in the Young group were Bacteroidetes, Actinobacteria, Fusobacteria, Firmicutes, Saccharibacteria, and Proteobacteria in turn. The phylum abundance ranking in the Elder group was Firmicutes, Bacteroidetes, Proteobacteria, Saccharibacteria, Actinobacteria, and Fusobacteria. And in the Older group, the abundance of Saccharibacteria replaced Proteobacteria as the third (Table S2).

Next, we evaluated the bacterial taxonomic composition at genus level. In this study, the relative abundances of the top 20 OP bacterial genera in each group were greater than 0.4% (Table S3). The most abundant genus in the OP microbiota of older elderly individuals was *Streptococcus*, followed by *unidentified_Saccharibacteria*, *Veillonella*, *unidentified_Prevotellaceae*, and *Neisseria*, all of which accounted for 63.87% of total abundance. The OP dominant genera of the younger elderly were *Streptococcus*, *Veillonella*, *Neisseria*, *unidentified_Prevotellaceae*, and *unidentified_Saccharibacteria*, which accounted for 62.77%. While the genera of young adults were *unidentified_Prevotellaceae*, *Fusobacterium*, *Alloprevotella*, *Rothia*, and *Actinomyces* in turn, all accounting for 63.98% (Table S3).

## Discrepancies of oropharyngeal microbiome in different age groups

According to the obtained effective sequences, OTUs were screened to determine the representative sequences, and bacterial diversity was determined based on OTUs. Finally, unweighted and weighted UniFrac distance matrix was used to analyze the bacterial community structure. The different richness indices (ACE and Chao1) indicated that the richness of OP microbiome in the Elderly group was significantly increased in comparison to the Young group (*P* = 0.0118 and 0.0316, respectively; Fig. 3A and B). For the diversity, two diversity indices (Shannon and Simpson) of OP microbiome in the Older group also were higher than the Young group (*P* = 0.0327 and 0.0068, respectively; Fig. 3C and D). No significant differences in richness and diversity were observed between the Elderly

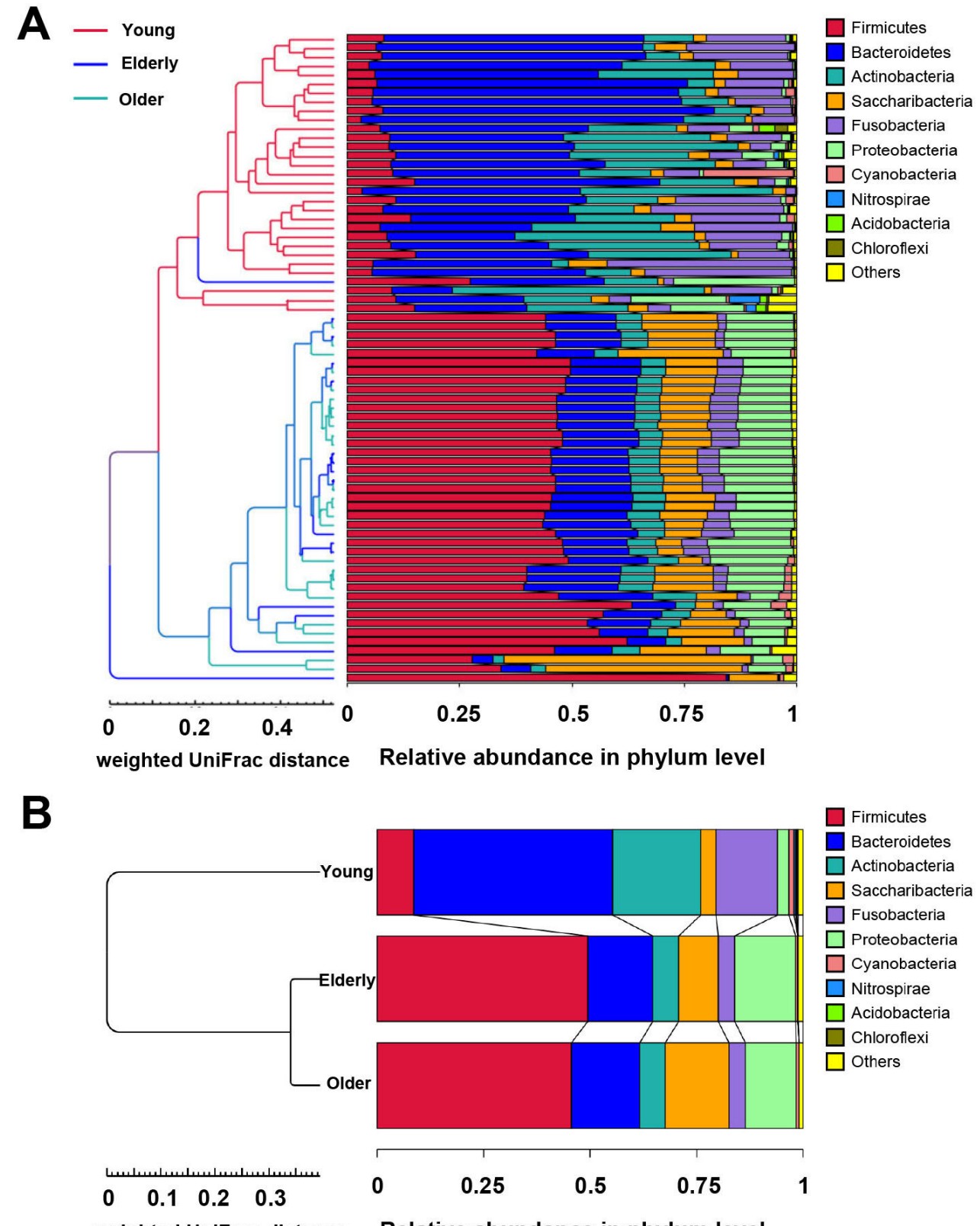

**FIG 2** Taxonomic profiles at phylum level of oropharyngeal microbiota in healthy groups. UPGMA cluster tree was based on weighted UniFrac distance in (A) each sample and (B) each group. Different color groups were used for each phylum.

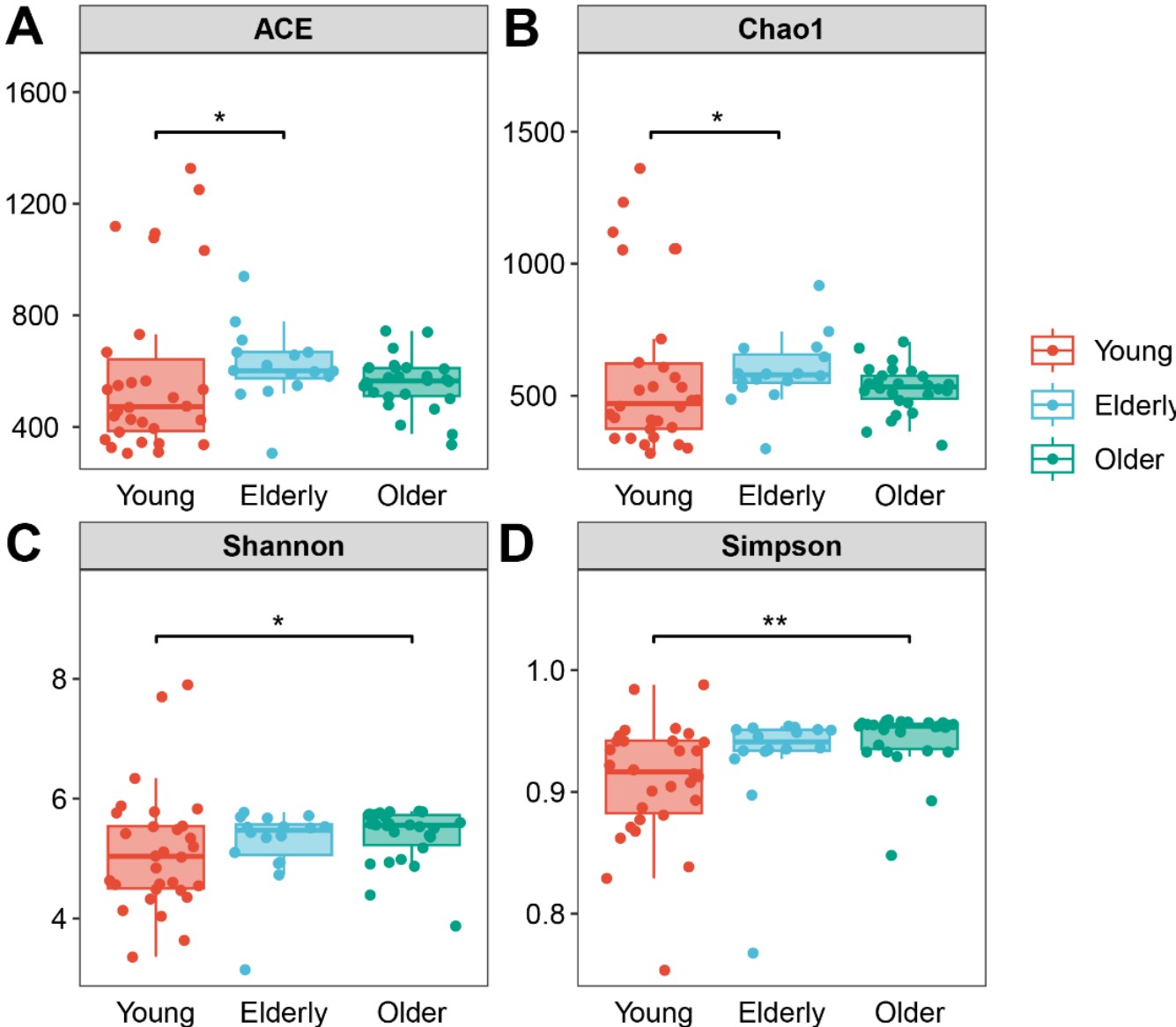

**FIG 3** The richness and diversity metrics of oropharyngeal microbiota in healthy groups. (A) Ace estimator, (B) Chao1 estimator, (C) Shannon index, and (D) Simpson index. Data are shown as box and whisker plots. The box indicates the interquartile range (IQR, 75th to 25th percentiles of the data), and the median value is shown as a line within the box; whiskers extend to the most extreme value within 1.5 × IQR, and outliers are shown as black dots. The results were analyzed by the non-parametric Kruskal-Wallis $H$ test followed by Dunn's multiple comparisons test and denoted as follows: *$P < 0.05$, **$P < 0.01$.

and Older groups. In short, these findings suggested that the species of oropharyngeal microbiota in the elderly are more diverse and uniformly distributed.

The unweighted (Fig. 4A) and weighted (Fig. 4B) UniFrac distance-based PCoA plots showed that microbial community structure in the Elderly and Older groups gathered into a similar cluster, while the Young group formed a separate cluster. The emerged pattern also illustrated the taxonomic differences among samples that were suggested in Fig. 2. Using an analysis of similarities (ANOSIM), we confirmed that oropharyngeal microbial community structure in the Young group was significantly different from that in the Elderly group (Fig. 4C, $R = 0.817$, $P = 0.001$) and the Older group (Fig. 4D, $R = 0.891$, $P = 0.001$). However, there was no significant difference in the microbial community structure between the Elderly and Older groups (Fig. 4E, $R = 0.062$, $P = 0.133$). Inter-group variance in microbial community structure displayed that the oropharynx of the elderly is colonized by different bacterial taxonomic compositions from the young.

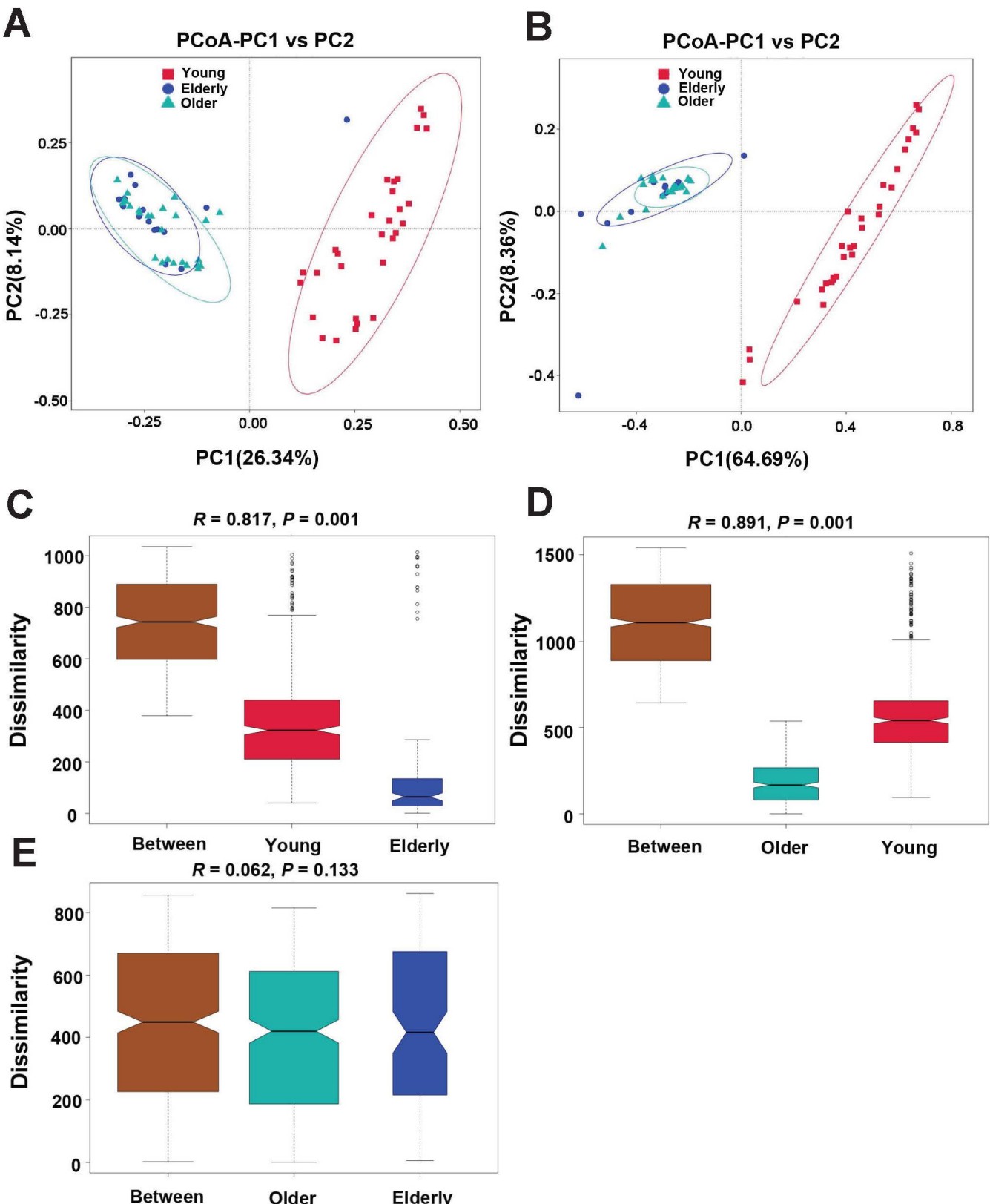

**FIG 4** The structure and between-group variance of oropharyngeal microbiota in healthy groups. (A) Unweighted and (B) weighted UniFrac distance-based PCoA. Each symbol represents a sample. The variance explained by the PCoA is indicated in parentheses on the axes (circles highlight the clustering of oropharyngeal microbiota in each group). Distance-dissimilarity between (C) Young and Elderly, (D) Young and Older, and (E) Older and Elderly was analyzed by ANOSIM analysis.

Metastats analysis at the phylum level showed that the relative abundances of Firmicutes, Saccharibacteria, and Proteobacteria in both the Elderly and Older groups were significantly enriched when compared to the Young group (Fig. 5A through C). Meanwhile, Bacteroidetes, Actinobacteria, and Fusobacteria were significantly lower (Fig. 5D through F). LEfSe analysis was used to compare the estimated OP microbiome phylotypes among healthy groups. The OP microbiome in the Young group was characterized by a preponderance of Bacteroidales, Prevotellaceae, Fusobacteriaceae, Micrococcaceae, Actinomycetaceae, Clostridiales, and Alphaproteobacteria. Whereas the Elderly group microbiome was dominated by Lactobacillales, Streptococcaceae, Gammaproteobacteria, Selenomonadales, Veillonellaceae, Neisseriaceae, and Pasteurellaceae. And the Older group contained predominately unidentified_Saccharibacteria (Fig. S2A and B).

The bacteria with relative abundance greater than 1% were identified as key genera. Compared to the Young group, eight key genera (*unidentified_Saccharibacteria*, *Oribacterium*, *Neisseria*, *Megasphaera*, *Streptococcus*, *Haemophilus*, *Veillonella*, and *Gemella*) in both the Elderly and Older groups showed a significant increase in abundance via Metastats analysis and were represented in a heatmap ($P < 0.01$) (Fig. 6). Conversely, seven key genera (*Leptotrichia*, *Rothia*, *Actinomyces*, *Alloprevotella*, *Fusobacterium*, *Atopobium*, and *unidentified_Prevotellaceae*) revealed a significant decrease in abundance ($P < 0.01$) (Fig. 6). In addition, in comparison with the Elderly group, *unidentified_Saccharibacteria* and *Campylobacter* were significantly increased, while *Actinobacillus* exhibited significantly decreased in the Older group ($P < 0.05$) (Fig. 6).

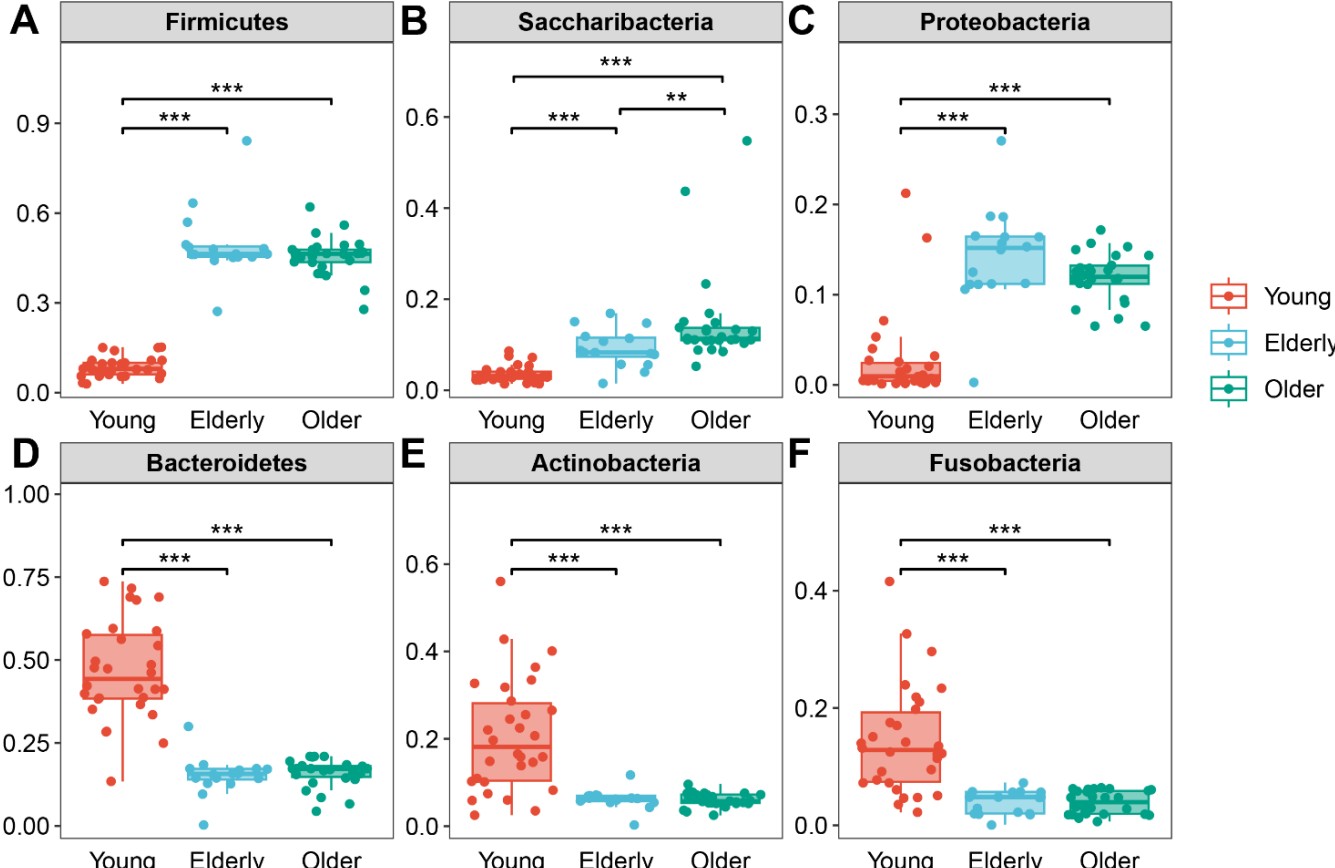

**FIG 5** Differences in relative abundance of dominant phylum of oropharyngeal microbiota in healthy groups. (A) Firmicutes, (B) Saccharibacteria, (C) Proteobacteria, (D) Bacteroidetes, (E) Actinobacteria, and (F) Fusobacteria. Data are shown as box and whisker plots. The box indicates the interquartile range (IQR, 75th to 25th percentiles of the data), and the median value is shown as a line within the box; whiskers extend to the most extreme value within 1.5 × IQR, and outliers are shown as black dots. The differences were analyzed by Metastats analysis and indicated by **$P < 0.01$, ***$P < 0.001$.

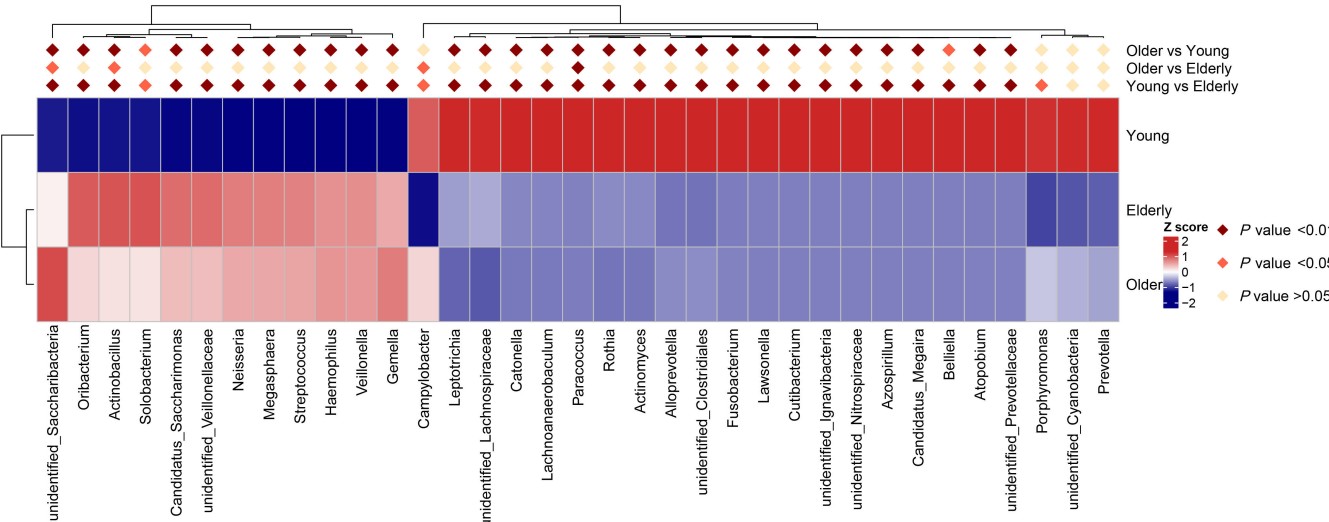

**FIG 6** Differences in relative abundance of dominant genera of oropharyngeal microbiota in healthy groups. The data were tested via Metastats analysis and represented in a heatmap for the abundant genera. The relative abundance of a genus in a given group is colored by its row *z*-score [(value–row mean)/row standard deviation]. Significant differences (*P* value) are denoted by rhombic dots shade.

## Correlation between age and oropharyngeal microbiota

When enrolled in all healthy subjects, the rose chart showed that the relative abundances of *Actinobacillus*, *Lachnoanaerobaculum*, *unidentified_Saccharibacteria*, and *Veillonella* were strongly correlated to age (Fig. S3A). By using canonical correspondence analysis, we found the strongly positive correlation effect of age on the relative abundances of *unidentified_Saccharibacteria*, *Veillonella*, and *Streptococcus*. Meanwhile, age had the largely negative correlation effect on *Lawsonella*, *Atopobium*, *Fusobacterium*, *Alloprevotella*, *unidentified_Prevotellaceae*, *Rothia*, and *Actinomyces* (Fig. S3B).

The observed age-related bacterial genera differences could be verified within healthy elderly populations over 65 years old due to the continuity of sampling age distribution. We confirmed a significantly positive correlation between the relative abundance of *unidentified_Saccharibacteria* and age ($r = 0.4412$, $P = 0.0034$). In contrast, the relative abundances of *Streptococcus*, *Oribacterium*, *Rothia*, and *Actinobacillus* decreased with age ($r = -0.4024$, $-0.4217$, $-0.3191$, and $-0.3181$; $P = 0.0083$, $0.0054$, $0.0394$, and $0.0401$, respectively) (Fig. 7).

## Similar microbial communities in heart failure patients and healthy controls

We assessed the impact of smoking (Fig. S4A), diabetes mellitus (Fig. S4B), hypertension (Fig. S4C), and heart failure grade III and IV according to the NYHA (Fig. S4D) on OP microbiome in the HF group and found that bacterial communities were not influenced by these physical statuses, as judged by the PCA.

No significant differences in richness (represented by ACE and Chao1) or diversity (represented by Shannon) between the H and HF groups were found (Table 2). However, the lower diversity (represented by Simpson) and slightly higher richness in the HF group indicated the disordered OP microbiome of heart failure patients possesses more richness but less evenness as compared to healthy controls. The microbial community structure in heart failure patients and healthy controls was also visualized by PCA (Fig. 8). The similar structure of microbial communities was found in the H and HF groups. A total of 98.3% of the sequences in the disease study team also belonged to the abovementioned six phyla. The relative abundances of dominating phyla remained remarkably constant in the elderly population with heart failure by Metastats analysis adjusted for multiple testing and false discovery rate (Table S4). The most abundant genus in the HF group was *Streptococcus*. An enigmatic unidentified bacterial genus,

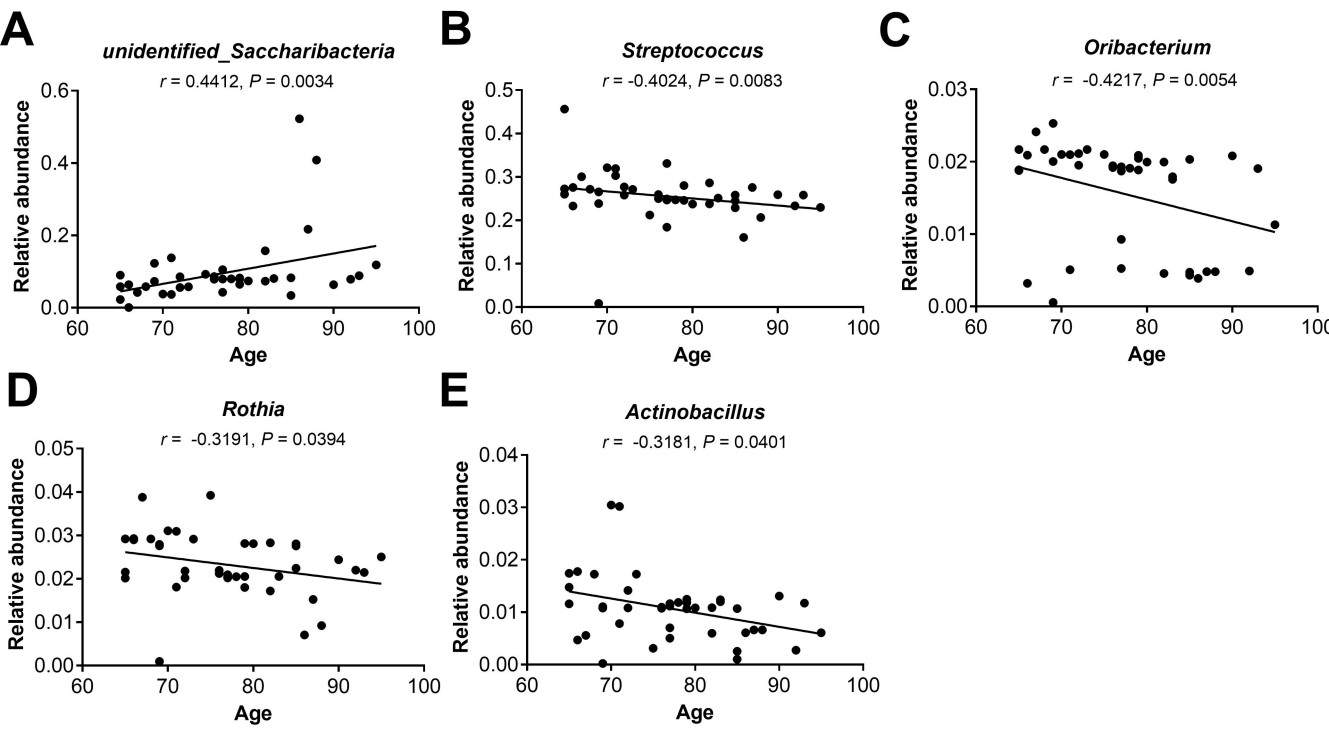

**FIG 7** Correlation between age and the relative abundance of oropharyngeal bacterial genera among healthy elderly subjects. These genera include (A) *unidentified_Saccharibacteria*, (B) *Streptococcus*, (C) *Oribacterium*, (D) *Rothia*, and (E) *Actinobacillus*. The data were analyzed by Spearman correlation analysis.

*unidentified_Saccharibacteria*, was also the abundant genus in both the H and HF groups. Within the abundant genera with relative abundance ≥1%, there were no significant changes, except *Gemella* and *Actinobacillus*, in the HF group when compared to the H group (Table S5).

## DISCUSSION

Currently, society is facing several major problems related to respiratory tract health. First, due to an aging population, primary, and secondary pneumonia in the elderly are more common than in any previous era (40). Second, respiratory infection results in high morbidity and mortality in the elderly with heart failure (41). Third, air pollution is another medical concern that is associated with an increased incidence of respiratory disease, particularly among the aged (42). A healthy respiratory tract, including commensal microbiota, should be the basis for preventing infection and dysbiosis. To date, limited information is available regarding the healthy OP microbiota, which is the first line of defense against microbes from ingested foods and inhaled air. Heilongjiang Province, located in North-East China, has long winters and the lowest winter temperatures at the same latitude http://www.tianqihoubao.com/lishi/. As such, it represents a cold geographic region with common cold-related diseases, such as respiratory tract infection (43) and cardiovascular disease (44). The healthy respiratory tract microbiota of

**TABLE 2** Comparison of alpha diversity indices of oropharyngeal microbiota between the H and HF groups[a]

| Index | H | HF | *P* value |
|---|---|---|---|
| ACE | 628.8911 ± 24.9912 | 664.5019 ± 27.8232 | 0.8017 |
| Chao1 | 600.2474 ± 25.5217 | 633.3581 ± 26.4971 | 0.5680 |
| Shannon | 5.4855 ± 0.0848 | 5.3953 ± 0.0902 | 0.1451 |
| Simpson | 0.9433 ± 0.0052 | 0.9324 ± 0.0066 | 0.0216 |

[a]Data are presented as the means ± SEM. H, health group; HF, heart failure group. Differences between the two groups were analyzed by Mann-Whitney *U* test.

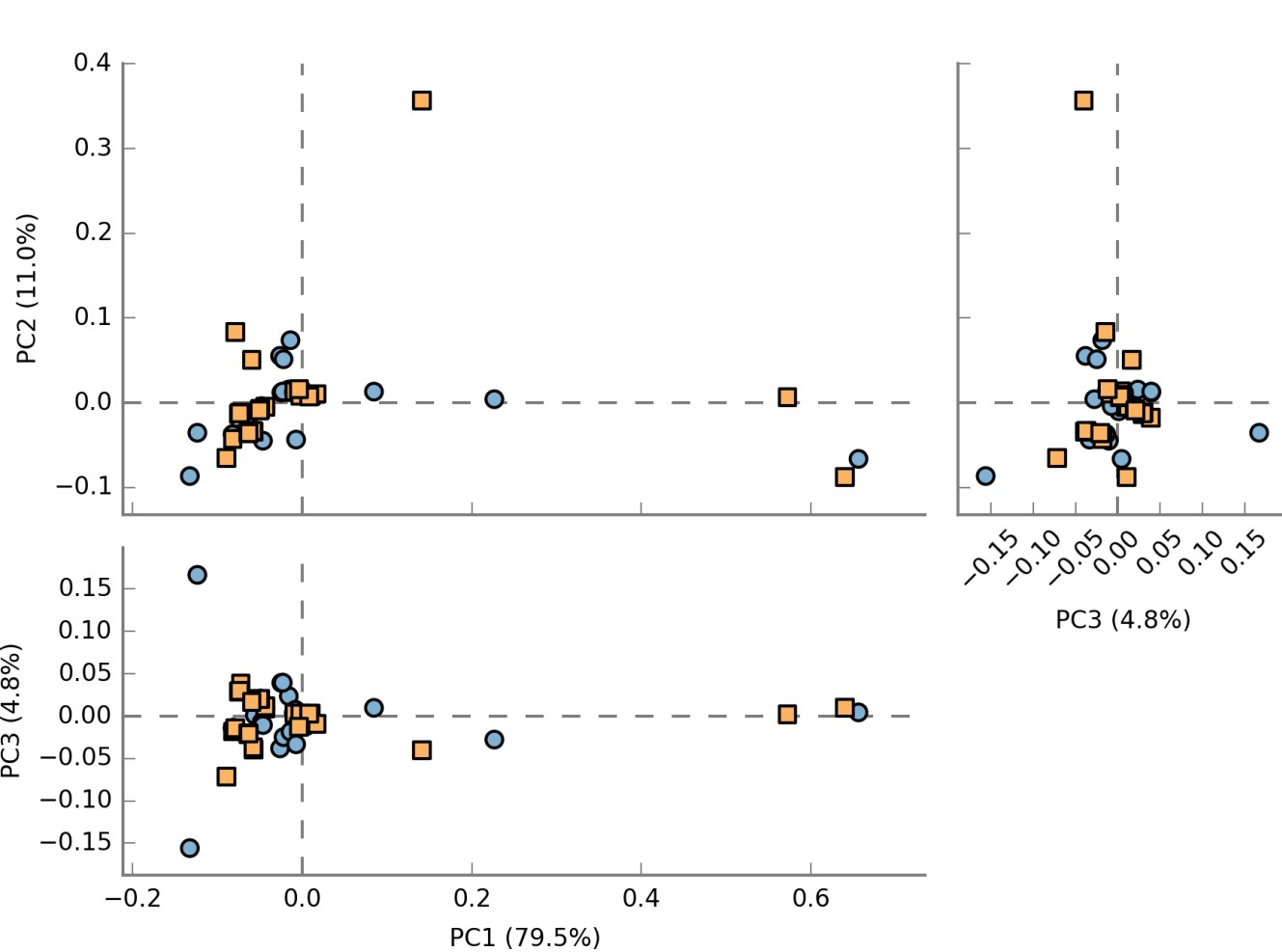

**FIG 8** The microbial community structure in heart failure patients and healthy controls was visualized by PCA. H, health group; HF, heart failure group.

the populations in this region will provide an important and unique context for respiratory disease prevention.

We chose to use high-throughput sequencing to type the OP microbiome in healthy people and the elderly with or without heart failure. The nasal, inguinal, and perianal regions were similar in microbial composition and significantly differed from the OP (45). It is generally assumed that the OP has its own characteristics throughout the respiratory tract. Some studies have confirmed bacterial communities within the OP were distinct from nostril (46–48), or NP (2, 47, 48) in healthy individuals. Although these nasal and OP communities in the elderly were not distinct from each other, each individual remained heterogeneous (26). Compared to other URT niches, the OP tended to have a small heterogeneity and a high proportion of *Streptococcus*, *Rothia*, *Prevotella*, *Gemella*, *Veillonella*, *Fusobacteria*, *Haemophilus*, and *Neisseria* (2, 47, 48). In general, the OP microbiome exhibited relatively stable patterns within a short time (49). A longitudinal study revealed that the OP microbiome of healthy adults, sampled weekly over 40 weeks, was characterized by high temporal stability (50). In addition, some findings indicated that the use of 12-week antiseptic mouthwash (51), or 48-week antibiotics (52) has minimal effects on the composition of the OP microbiota. And a previous study showed that OP bacterial profiles were significantly related to the age of the subjects (11). Hence, we looked at the OP microbiome in different age groups, and then the elderly and young adults exhibited a distinct OP bacterial community structure.

The microbiome research focused on the dynamics and development of OP microbiota. The neonates harbored bacterial communities that were undifferentiated across multiple body habitats and depended on their delivery mode (53). The vaginally delivered infants acquired bacterial communities similar to their mother's vaginal microbiota, dominated by *Lactobacillus*, *Prevotella*, and *Sneathia*. The bacterial communities obtained by C-section babies were similar to those on the skin surface, made up mainly of *Staphylococcus*, *Corynebacterium*, and *Propionibacterium* (53). In the healthy newborns, breastfeeding correlated with changes in the OP microbiota composition and had the greatest impact upon the relative abundance of *Streptococcus* and *Candida* (54). The OP microbiota in 6-week-old infants was already dominated by *Streptococcus*, accounting for more than 70%, followed by *Veillonella*, *Gemella*, and *Haemophilus* with a steady proportion. Other common OP genera *Neisseria*, *Prevotella*, and *Alloprevotella* increased to a stable presence at 6 months of age (55). In a study on OP microbiome of children aged 0.1–10.8 years, *Streptococcus* was the most common in children aged ≤1 years (52.7%), then decreased to 25.7% (>1 and ≤3 years) and 19.5% (>3 years), accompanied by an increase in the abundance of the four major OP genera *Neisseria*, *Haemophilus*, *Leptotrichia*, and *Prevotella* (47). In short, the OP microbiota in children was consistently characterized by a high prevalence of the genus *Streptococcus* which showed a decreasing trend along with an increasing age. Compared to adults, OP microbial communities in children (2) or elderly (26) had a significantly higher proportion of *Streptococcus*. Interestingly, a literature reported that in healthy young populations from Northeastern China as well, the genus *Prevotella* (a member of Prevotellaceae) has the highest relative abundance, while *Streptococcus* has not been observed with high abundance (56). A similar situation also exists in our study. The level of *unidentified_Prevotellaceae* was the highest in the young adult, while *Streptococcus* was lower. The negative association between the relative abundance of *Streptococcus* and age was found in the elderly. These results greatly reflected the dynamics of *Streptococcus* in OP colonization of healthy human beings. After reaching the peak of colonization in early infancy, *Streptococcus* continues to decline until adulthood and recovers to a certain high proportion in the elderly.

In this study, the relative abundance of *unidentified_Saccharibacteria* increased with age among three age groups. Bacteria from the Saccharibacteria phylum are ubiquitous members of the human oral microbiome and are part of the candidate phyla radiation. Both studies also observed a significant proportion of Saccharibacteria in the OP (10, 56). In addition, Saccharibacteria was strongly increased in negative controls and SARS-CoV-2 paucisymptomatic patients as compared to SARS-CoV-2 ICU patients (14). Nevertheless, due to the lack of cultivated representatives, knowledge of Saccharibacteria was scarce. At present, major breakthroughs have been made in the cultivation and characterization of Saccharibacteria (57, 58). Accumulating evidence showed that members of the Saccharibacteria phylum may play a role in periodontal disease (59). We did not claim that the increased abundance of *unidentified_Saccharibacteria* is a healthy marker for long-lived individuals in this region. Because these species may benefit from the local environment and are associated with overuse of the oral or tooth in the elderly. However, further research, to characterize these species and explore their significance in healthy aging, is warranted.

In our study, OP bacterial diversity and taxonomic composition were not significantly different between heart failure patients and healthy controls. The OP microbiome is resilient in disease, and the stable microbiome is important in disease. This robustness of the OP microbiome has been confirmed by other studies. The diversity of bacterial communities remained remarkably stable following the acquisition of influenza, with no significant differences over time between individuals with influenza and controls (60). The comparison of OP microbiota between healthy children and children with asthma or cystic fibrosis revealed there was a similar and core microbiota represented by *Prevotella*, *Streptococcus*, *Neisseria*, *Veillonella*, and *Haemophilus* (16). Santiago et al. confirmed the absence of significant differences in people with and without severe

asthma (61). Specifically, Streptococcus-dominated microbiota was enriched in recovered patients, and showed high temporal stability (20). Future microbiome-targeted therapies for respiratory disease prevention in the elderly, based on the native stable OP microbiome, are worthy of investigation.

Human beings live in different regions, and the geographical environment will directly affect the distribution of human microbiota. Therefore, it is very important to characterize the OP microbiome in specific geographical contexts. The abundance of certain bacterial genera varies in geographic regions, even though the microbial composition may be similar. For example, the adult OP microbiome was dominated by *Streptococcus* (26.1%), *Prevotella* (14.1%), and *Veillonella* (8.9%) in Ontario, Canada (26). In Calgary, Canada (2), America (48), and Korea (11), the adults were likewise dominated by the above three genera. However, the relative abundance of *Streptococcus* increased to more than 50% in the total bacterial community. Recently, Lu et al. found that the OP microbiota was predominantly enriched with *Prevotella* (21.15%), *Neisseria* (17.74%), *Porphyromonas* (8.40%), *Haemophilus* (7.92%), *Veillonella* (6.87%), and *Streptococcus* (6.40%), which differs from the findings reported herein (10). Possible reasons for these differences may be due to the fact that the subjects come from southern China. Each living environment includes specific air exposure and eating habits that could influence members of the OP microbiota. For instance, during the winter heating period in Northeast China, the imbalance of the OP microbiota might be caused by air pollution (56). Studies about homogeneous populations found significant differences in OP microbial diversity and composition between Australian Chinese and China-born children, or newly-arrived and long-term Chinese immigrants (62, 63). Environmental factors may cause changes in the OP microbiota, which may be a potential risk factor for allergy-related diseases. These results reflect the shaping power of environment or lifestyle on OP microbiota. It is necessary to control for region-related variables when studying other factors on OP microbiota. To establish microbiome databases for different regions is vital.

A limitation of this study was the relatively high number of heart failure patients with one or more comorbidities, including smoking. However, we assessed the independent effect of comorbidities on the OP microbial composition in heart failure patients. This is consistent with previous research in which the microbiota of the elderly showed no associations with sex, comorbidities, residence, or vaccinations (26). Another potential limitation was the concentrated age distribution of young adult individuals, which may contribute to the absence of mid-aged adults and form a discontinuous layer. Further, it is important to note that the cross-sectional design of the study possesses a major limitation as causality cannot be assessed. For this study, we recognize that the inclusion of samples from low-latitude regions as controls will provide a better understanding of the core microbiome characterized by high-latitude and cold regions. Next-generation sequencing studies reveal valuable microbiota information associated with specific human niches. The second step requires us to re-emphasize the extensive culture and identification of bacteria, although this is the first method to study microorganisms, which is consistent with the recently proposed culturomics (64). Moreover, the use of isolated bacteria is no longer limited to individual strains, and the establishment of synthetic microbial communities will play a greater role. How to utilize the core microbiome, microbial interaction network and extensive culture to study the beneficial effects of synthetic microbial communities in the respiratory tract will be an important direction for future research.

In conclusion, the OP microbiome in the elderly showed a markedly different microbial community than in the young, such as bacterial diversity, structure, and abundance. *Streptococcus* was the most prevalent genus in the OP microbiome of elderly people, whereas *unidentified_Prevotellaceae* dominated in young adults. The evidence of *Streptococcus* competing with members of Prevotellaceae for niche dominance was further validated. The OP core microbiota of the elderly in high-latitude and cold regions was identified by a high abundance of *Streptococcus*, *unidentified_Saccharibacteria*,

*Veillonella*, *unidentified_Prevotellaceae*, *Neisseria*, and *Alloprevotella*. In OP core microbiota, more attention should be paid to colonization dynamics of *Streptococcus* and members of Saccharibacteria in old age and even in life. In addition, the relationship between OP microbiota and heart failure was sought. Although no significant difference in the OP microbiome was found between heart failure and healthy elderly, the lower diversity indicated the potential OP microbial disorder in heart failure patients. The robustness and function of the OP microbiome could be highly valued.

## ACKNOWLEDGMENTS

This work was supported by Grant for Innovation Team Program of Heilongjiang Province to H.L., the National Science Foundation of China to X.-H.Y. (81570437), the Shenzhen Science and Technology ProgramProject to X.-H.Y. (General project, JCYJ20220531102612027), and the Shenzhen Excellent Science and Technology Innovation Talent Training Project to K.-L.Y. (Doctoral initiation program, RCBS20221008093102010).

H.L. and X-H.Y. conducted and organized the study. H.L. designed the experiment. X.-Y.H., Y.Z., and W.-J.C. involved in collection of the samples and patient data. J.L., X.-Y.H. performed the experiments, data analysis, and wrote the original draft. J.L. and E.-Y.D. involved in bioinformatics analysis and graphical representation. A.K.R. partially involved in writing and language modification. H.L, X.-H.Y. and K.-L.Y. corrected the draft and approved the final version of the manuscript. D.L. and M.Z. involved in project administration. H.L., X.-H.Y. and K.-L.Y. were responsible for funding acquisition. All authors involved in the manuscript preparation and approved the final version of the manuscript.

## AUTHOR AFFILIATIONS

[1]Department of Microbiology, Harbin Medical University, Harbin, China
[2]Department of Endocrinology and Metabolism, Shenzhen University General Hospital, Shenzhen, China
[3]Department of Cardiology, First Affiliated Hospital of Harbin Medical University, Harbin, China
[4]Wu Lien-Teh Institute, Harbin Medical University, Harbin, China
[5]Heilongjiang Provincial Key Laboratory of Infection and Immunity, Harbin, China
[6]Department of Cardiology, Shenzhen University General Hospital, Shenzhen, China

## AUTHOR ORCIDs

Jian Liu (iD) http://orcid.org/0000-0003-1615-3389
Xin-Hua Yin (iD) http://orcid.org/0000-0002-5874-1448
Hong Ling (iD) http://orcid.org/0000-0002-1886-1242

## FUNDING

| Funder | Grant(s) | Author(s) |
| --- | --- | --- |
| Grant for Innovation Team Program of Heilongjiang Province | | Hong Ling |
| National Science Foundation of China | 81570437 | Xin-Hua Yin |
| Shenzhen Science and Technology Program Project | JCYJ20220531102612027 | Xin-Hua Yin |
| Shenzhen excellent science and technology innovation talent training project | RCBS20221008093102010 | Ke-Laier Yang |

## AUTHOR CONTRIBUTIONS

Jian Liu, Data curation, Formal analysis, Methodology, Visualization, Writing – original draft | Xiao-Yu He, Data curation, Formal analysis, Methodology, Writing – original draft

| Ke-Laier Yang, Funding acquisition, Writing – review and editing | Yue Zhao, Data curation | En-Yu Dai, Formal analysis, Visualization | Wen-Jia Chen, Data curation | Aditya Kumar Raj, Writing – original draft, Writing – review and editing | Di Li, Project administration | Min Zhuang, Project administration | Xin-Hua Yin, Conceptualization, Funding acquisition, Project administration, Writing – review and editing | Hong Ling, Conceptualization, Data curation, Formal analysis, Funding acquisition, Project administration, Supervision, Validation, Writing – review and editing

## DATA AVAILABILITY

The raw sequencing data supporting the conclusions of this article will be made available by the authors, without undue reservation. All raw sequencing reads are available at NCBI Sequence Read Archive (SRA) database (PRJNA1100324) .

## ADDITIONAL FILES

The following material is available online.

### Supplemental Material

**Supplemental material (Spectrum00216-24-s0001.docx).** Tables S1 to S5; Fig. S1 to S4.

### Open Peer Review

**PEER REVIEW HISTORY (review-history.pdf).** An accounting of the reviewer comments and feedback.

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
