## [Reviewer comments · Microbiology Spectrum]

Microbiology Spectrum

Oropharyngeal microbiome profiling and its association with age and heart failure in the elderly population from the northernmost province of China

Jian Liu, Xiao-Yu He, Ke-Laier Yang, Yue Zhao, En-Yu Dai, Wen-Jia Chen, Aditya Raj, Di Li, Min Zhuang, Xin-Hua Yin, and Hong Ling

Corresponding Author(s): Hong Ling, Harbin Medical University

Review Timeline:

Submission Date:	January 23, 2024
Editorial Decision:	March 29, 2024
Revision Received:	May 10, 2024
Accepted:	July 7, 2024

Editor: Zhe LYU

Reviewer(s): Disclosure of reviewer identity is with reference to reviewer comments included in decision letter(s). The following individuals involved in review of your submission have agreed to reveal their identity: Baohong Wang (Reviewer #2); Lili Ren (Reviewer #5)

Transaction Report:

DOI: <https://doi.org/10.1128/spectrum.00216-24>

Re: Spectrum00216-24 (The Altered Oropharyngeal Microbiome in the Elderly Is Associated with Age, but Not Heart Failure Disorder)

Dear Prof. Hong Ling:

Thank you for the privilege of reviewing your work. Below you will find my comments, instructions from the Spectrum editorial office, and the reviewer comments.

In addition, please note that the language of your manuscript contain significant overlapping texts from published literature. Specifically, the introduction, materials and methods, and statistical language. Please revise accordingly, especially the first 3 paragraphs of the introduction section.

Revision Guidelines

Sincerely,
Zhe LYU
Editor
Microbiology Spectrum

Reviewer #2 (Comments for the Author):

Respiratory tract infections are the most common triggers for heart failure in elderly people. Here, Jian Liu et al. investigated the oropharyngeal (OP) microbiome in the elderly population in Heilong Province in the North-East of China, who located in high-

latitude and cold area with a high prevalence of respiratory tract infection and heart failure. They characterized the intestinal microbiota in the elderly people when compared with those in young adults. In addition, they reported the correlation between the age and some specific microbes, but did not find the specific intestinal microbes in patients with heart failure. Thus, these authors considered that they identified the respiratory tract core microbiota associated with the high latitude and cold regions, and revealed the robustness of OP microbiome in the aged population, which could supply the basis for microbiome-target interventions. The findings of the study have clinical significance and interesting. However, the present experiment data could not provide enough evidence for the conclusion and more confirmation improvements are required.

It is well-known that there is significant different diet pattern between people lived in north and south China. If these authors aimed to strengthen the potential influence of latitude/ cold regions on gut microbiome in their study, the effect of diet on gut microbiome should be considered.

The study should be concluded with the evidence in the study. The reviewer suggested that these authors firstly should summarize the novelty findings in their findings. Then, the following questions should be considered and answered. For example, were there significant connection between the respiratory tract core microbiota and high latitude/ cold regions in the metagenomics study? Were there the specific microbes identified in the metagenomics study in patients with heart failure? Studies should be carried out to strengthen the cause-effect relationship between gut microbiota (specific microbes in aged people), and heart failure; as well as the respiratory tract infection and heart failure.

The title should be revised. Also, the case of the letters in the title was not written correctly.

Reviewer #5 (Comments for the Author):

In this study, the authors described the composition of microbiome in the elderly based on 42 individuals aged {greater than or equal to} 65 years using 16S rRNA sequencing, and the oral microbiome differences between young and old people were compared by including 30 college students. The association of microbiome with heart failure in older adults was analyzed. The profiles of higher diversity of microbiota and dominant phyla in the OP microbial in elderly people revealed age-related oral microbiome and microbiome-targeted interventions. The findings would provide much more evidence to understand the profiles of oral microbiome in high latitude and cold area. Several questions are listed as followed.

1. As of the inclusion and exclusion criteria of the study, please clarify the detail information, especial the "healthy elderly people" mentioned in the study. Do these 42 elderly people have underlying diseases? Table1 shows that some people have diabetes and hypertension. Whether they have other underlying diseases? What is the standard document for "healthy elderly people"?
2. The study tended to describe different profile of the microbiome in different age groups. However, the age groups were decided only in 22-26 years old and 65-95 years old, Please give appropriate reasons?
3. A total of 42 elderly people were included in the sample. Whether the small size of samples represent the elderly population?
4. In Table 1, a large proportion of the elderly with heart failure are smokers (13/32), while none of the elderly without heart failure are smokers. Previous studies have shown that smoking has an impact on the microbiome, but the microbiome has nothing to do with heart failure in the conclusion of the paper. How the author explains the impact of smoking on the microbiome?
5. In Figure 3, the diversity of the microbiome is significantly higher in the elderly than in college students aged 22-26 years. Please provide other literature evidence to support this? Previous study has shown that communities with higher diversity are more stable. How does this conclusion be supported?

Specific comments:

- 1) In Result section (Line 218), it is mentioned that two cohort were established, including "age cohort" and "disease cohort". Is there longitudinal follow-up in the study? If not, please change it in the whole text.
- 2) When comparing the association between heart failure and microbiome in the elderly, the author made age matching in the control group, but the gender difference was large (Table1). If the conclusion is established, please supplement the evidence supporting that gender has no effect on the microbiome in the elderly.

Point Response Letter

Microbiology Spectrum

Dear Editor Zhe LYU,

Thank you for your decision letter concerning our manuscript (ID Spectrum00216-24) entitled "The altered oropharyngeal microbiome in the elderly is associated with age, but not heart failure disorder", and your time regarding for our revision. I also appreciate all the critical comments from you and reviewers. We have carefully considered the comments and revised the manuscript accordingly. With these improvements, we hope that the current version can meet the Journal's standards for publication. The following is a point-by-point response to all those comments and a list of changes we have made to the manuscript.

Sincerely

Hong Ling

Point-by-point responses to the comments of the Editor and reviewers, and a list of changes are:

(The comments of the editor and reviewers are in italics and blue color, which are followed by our responses.)

Editor

Please note that the language of your manuscript contain significant overlapping texts from published literature. Specifically, the introduction, materials and methods, and statistical language. Please revise accordingly, especially the first 3 paragraphs of the introduction section.

A: We thank you for pointing out the language problem.

We have carefully checked through the whole manuscript and emphatically renewed the first 3 paragraphs of the introduction section and part of the materials and methods as the editor suggested. The revised text has been highlighted.

Data availability: *ASM policy requires that data be available to the public upon online posting*

of the article, so please verify all links to sequence records, if present, and make sure that each number retrieves the full record of the data. If a new accession number is not linked or a link is broken, provide Spectrum production staff with the correct URL for the record. If the accession numbers for new data are not publicly accessible before the expected online posting of the article, publication may be delayed; please contact production staff (Spectrum@asmusa.org) immediately with the expected release date.

To meet the requirement that the data is available to the public, we have posted all raw sequencing reads in NCBI Sequence Read Archive (SRA) database (PRJNA1100324) (<https://www.ncbi.nlm.nih.gov/sra/PRJNA1100324>). We have also added *data availability statement* in the revised manuscript.

Reviewer: 2

Comments to the Author

Respiratory tract infections are the most common triggers for heart failure in elderly people. Here, Jian Liu et al. investigated the oropharyngeal (OP) microbiome in the elderly population in Heilong Province in the North-East of China, who located in high-latitude and cold area with a high prevalence of respiratory tract infection and heart failure. They characterized the intestinal microbiota in the elderly people when compared with those in young adults. In addition, they reported the correlation between the age and some specific microbes, but did not find the specific intestinal microbes in patients with heart failure. Thus, these authors considered that they identified the respiratory tract core microbiota associated with the high latitude and cold regions, and revealed the robustness of OP microbiome in the aged population, which could supply the basis for microbiome-target interventions. The findings of the study have clinical significance and interesting. However, the present experiment data could not provide enough evidence for the conclusion and more confirmation improvements are required.

*It is well-known that there is significant **different diet pattern** between people lived in north and south China. If these authors aimed to strengthen the potential influence of latitude/ cold regions on gut microbiome in their study, **the effect of diet on gut microbiome should be considered.***

A: We fully agree with the reviewer.

The effects of diet on gut microbiome and on OP microbiome should be considered. There

are numerous references regarding the association of diet and gut microbiome. However, there are very limited literature regarding the association and diet and respiratory microbiome. Guo et al have documented the significant differences in the diversity and structure of oropharyngeal microbiota between Han Chinese children living in Australia and China, or between newly-arrived and long-term Chinese immigrants (**World Allergy Organ J. 2019 Aug 9;12(8):100051; Allergy Asthma Clin Immunol. 2020 Jul 25;16:67**). The study further suggests that the Western environment/lifestyle has created a different oropharyngeal microbiome.

We acknowledge that the inclusion of southern Chinese population as a control and considering dietary factors can much better show the characteristics of oropharyngeal microbiome in high latitude/cold regions. It is significant to do further cohort-based study involving the diet influence consideration. We have added the description regarding this limitation in the discussion section (**Page 12, Lines 477**).

The study should be concluded with the evidence in the study. The reviewer suggested that these authors firstly should summarize the novelty findings in their findings. Then, the following questions should be considered and answered.

A: We are grateful for this suggestion.

The evidence and novelty of the findings in this study have been summarized in the last paragraph of the discussion section. The abstract section has been revised accordingly.

Specific comments:

1. For example, were there significant connection between the respiratory tract core microbiota and high latitude/ cold regions in the metagenomics study?

A: We thank the reviewer for the comment.

In this study, we have identified the respiratory tract core microbiota of the elderly in high-latitude/cold regions, characterizing by a high abundance of *Streptococcus*, *Saccharibacteria*, *Veillonella*, *Prevotella*, *Neisseria* and *Alloprevotella*. This study provides the baseline data of characters of OP microbiome with age-associated manner in a high-latitude region.

The oropharyngeal microbiota of healthy individuals, found in Hangzhou, China (**Emerg Microbes Infect. 2017 Dec 20;6(12):e112**), is different from the results of the present study. The reason for this difference may be that the subjects live in different regions, where the climate differences are obvious. These results indicate that the oropharyngeal microbiota has

regional characteristics. We propose that each living environment contains specific air exposures and dietary habits that may influence members of the microbiota in the oropharynx. In order to determine whether the core microbiota of the respiratory tract is associated with high-latitude, our current evidence is indeed insufficient. To this end, some studies involving cohorts from different geographical regions, high-, middle- and low-latitude, with large sample number should be conducted.

2. Were there the specific microbes identified in the metagenomics study in patients with heart failure? Studies should be carried out to strengthen the cause-effect relationship between gut microbiota (specific microbes in aged people), and heart failure; as well as the respiratory tract infection and heart failure.

A: We thank the reviewer for the suggestions.

We did compare the core microbiota between healthy and heart failure elderly. There were no significant changes, except *Gemella* and *Actinobacillus*, in core microbiota (**Table S5**). Unfortunately, we did not find indicator microbes that could distinguish heart failure from healthy elderly. But the lower diversity (**Table 2**-Simpson index) and slightly higher richness (ACE and Chao1 index) indicated the potential OP microbial disorder in heart failure patients. This may be because heart failure is indeed a weak factor affecting the oropharyngeal microbiome relative to other factors such as climate. In this study, we found some important phenomena of the existence of specific microbes in the elderly, such as the competition between *Streptococcus* and members of Prevotellaceae for niche dominance and the oropharyngeal colonization dynamics of *Streptococcus* and members of Saccharibacteria (see **paragraphs 3 and 4 of the discussion** for details).

Cross-sectional studies possess a major limitation as causality cannot be assessed. To the best of our knowledge, currently, research on the causal relationship between microbiota and disease has focused on the gut. There is a lack of report on the causal relationship between the respiratory microbiota and disease. In the future, appropriate models and longitudinal studies in animal experiments will be needed to verify the impact of specific microorganisms on heart failure or other diseases.

Here, we did not have direct data on the association between respiratory infections and heart. Some studies have shown a strong association between heart failure and respiratory infections. Further studies are still needed to investigate the causal relationship between respiratory infections and heart failure.

3. The title should be revised. Also, the case of the letters in the title was not written correctly.

A: We thank the Reviewer and apologize for the inaccuracy.

We have updated new title as “Oropharyngeal microbiome profiling and its association with age and heart failure in the elderly population from the northernmost province of China”.

Reviewer: 5

Comments to the Author

In this study, the authors described the composition of microbiome in the elderly based on 42 individuals aged {greater than or equal to} 65 years using 16S rRNA sequencing, and the oral microbiome differences between young and old people were compared by including 30 college students. The association of microbiome with heart failure in older adults was analyzed. The profiles of higher diversity of microbiota and dominant phyla in the OP microbial in elderly people revealed age-related oral microbiome and microbiome-targeted interventions. The findings would provide much more evidence to understand the profiles of oral microbiome in high latitude and cold area. Several questions are listed as followed.

Major comments:

1. As of the inclusion and exclusion criteria of the study, please clarify the detail information, especial the "healthy elderly people" mentioned in the study. Do these 42 elderly people have underlying diseases? Table1 shows that some people have diabetes and hypertension. Whether they have other underlying diseases? What is the standard document for "healthy elderly people"?

A: We thank the Reviewer for this important suggestion.

Following the Reviewer’s comment, we have elucidated more detailed inclusion and exclusion criteria. Healthy elderly people were eligible if: a) no use of antibiotics, antifungal drugs, antiviral drugs, antiparasitic drugs, corticosteroids, cytokines, antineoplastic drugs, immunosuppressive drugs and probiotics in the past 3 months; b) age ≥ 65 years old; c) never smoking. Exclusion criteria for healthy elderly people were as follows: a) oropharyngeal deformities or tumors, underlying diseases such as asthma; pharyngitis, rhinitis, sinusitis, etc; b) transient syncope in the past 3 months; c) symptoms that affect breathing such as heart failure; d) influenza vaccination in the last 3 months; e) previous history of infectious diseases such as tuberculosis, hepatitis B, and hepatitis C; f) acute illness with or without fever in the past 3 months. Accordingly, we have modified the text in the revised manuscript (**Page 5, Lines 162**).

According to the criteria for healthy elderly in China issued by the National Health Commission of the People's Republic of China, these people should meet the following requirements: a) self-care or basic self-care; b) organ changes with aging not leading to significant functional abnormalities; c) controlling risk factors for health within age-appropriate limits; d) good nutritional status; e) basically normal cognitive function; f) optimistic and positive with self-satisfaction; g) certain health literacy and a good lifestyle; h) active participation in family and social activities; i) good social adaptability.

The enrolled elderly individuals in the study had no underlying respiratory organic or infectious diseases other than diabetes and hypertension. We considered that diabetes and hypertension occur mainly in the elderly over 75 years of age, and these health risk factors are controlled within age-appropriate limits. In addition, we assessed the independent effect of comorbidities on the OP microbial structure in heart failure patients in **Figure S4**. Results obtained were consistent with previous research in which the microbiota of the elderly showed no associations with comorbidities (**Ann Am Thorac Soc. 2014 May;11(4):513-21**).

2. The study tended to describe different profile of the microbiome in different age groups. However, the age groups were decided only in 22-26 years old and 65-95 years old. Please give appropriate reasons?

A: We thank the Reviewer for pointing this out.

The original idea of present study was to focus on the profile of OP microbiome in elderly people to find significant characters for future intervention in both health and disease condition. Therefore, we involved age group of the older than 75-year. The elderly population is defined as people aged 65 and over (see <https://data.oecd.org/pop/elderly-population.htm#indicator-chart>). The cut-off of 75-year-old between younger and older elderly was consistent with that in previous studies (**Diabetes Metab Res Rev. 2015 Feb;31(2):204-11**). We selected the Young group's samples from healthy college students during the physical examination in affiliated hospitals as the control. Therefore, the age range was very narrow. The concentrated age distribution of young adult individuals and the absence of mid-aged adults meet our goals. However, to clarify the age-associated tendency of OP microbiome, we need conduct the studies involving discontinuous age layers.

3. A total of 42 elderly people were included in the sample. Whether the small size of samples represent the elderly population?

A: We are very grateful to the Reviewer for the critical question.

We fully agree that a large number of samples better represent the respiratory microbiome profile of older adults. Our efforts to collect a large number of samples were difficult after removing samples that did not meet inclusion criteria. In a cross-sectional study, a certain number of samples can explain the findings to a certain extent. In a recent study, Ma et al. presented the oropharyngeal microbiota characteristics identified by metagenomic sequencing analyses of oropharynx swab specimens from 31 COVID-19 patients and 28 healthy controls and revealed a distinct oropharyngeal microbiota composition in the COVID-19 patients (**Signal Transduct Target Ther. 2021 May 13;6(1):191**).

3. In Table 1, a large proportion of the elderly with heart failure are smokers (13/32), while none of the elderly without heart failure are smokers. Previous studies have shown that smoking has an impact on the microbiome, but the microbiome has nothing to do with heart failure in the conclusion of the paper. How the author explains the impact of smoking on the microbiome?

A: We thank the reviewer for the comment.

In this study, we firstly evaluated the impact of smoking on the microbiome using samples from Heart-failure group and found that bacterial communities were not influenced by this status (**Figure S4**).

Indeed, most studies have described that smoking associated with alterations of microbiome in different body sites in health and diseases. But there are exceptions to the current evidence. Erb-Downward et al. showed that the diversity of bacterial communities in bronchoalveolar lavage fluid (BALF) from healthy smokers was similar to that from healthy never-smokers, and COPD patients; they also found the extensive overlap in the bacterial community membership between among the three groups (**PLoS One. 2011 Feb 22;6(2):e16384**). Another study failed to demonstrate the association between smoking and diversity of bacteria, because of no difference in bacterial dominance from induced sputum in asthma patients between ex-smokers and non-smokers (**PLoS One. 2014 Jun 23;9(6):e100645**).

The mechanisms underlying the influence of smoking on the microbiome are complicated and may include changing immune homeostasis, biofilm formation, oxygen tension, or through direct contact of microbes with the shaped environment of chemicals, heavy metals, particulate matter and other constituents in tobacco. But the microbiome in a human body has the stability and resilience to restore themselves after perturbation maintains homeostasis in health. For example, in a longitudinal study of a type A H3N2 influenza virus intranasal inoculation

experiment, the diversity of oropharyngeal bacterial communities remained stable after influenza infection, with only slight changes in the microbiome (**Clin Infect Dis. 2019 May 30;68(12):1993-2002**). A comparison of the oropharyngeal microbiome between healthy children and children with asthma or cystic fibrosis showed a high degree of similarity (**Mediators Inflamm. 2017;2017:5047403**). Santiago et al. also demonstrated no significant difference between the oropharyngeal microbiome of people with and without severe asthma (**BMC Microbiol. 2017 May 10;17(1):109**).

Perhaps, the respiratory microbiome is influenced by various factors, and smoking is only a weak factor under some dominant factors such as climate. Longitudinal studies integrating metagenomic, transcriptomic, metabolomic methods with clinical results may help to ascertain the relationships between smoking and microbiome.

5. In Figure 3, the diversity of the microbiome is significantly higher in the elderly than in college students aged 22-26 years. Please provide other literature evidence to support this? Previous study has shown that communities with higher diversity are more stable. How does this conclusion be supported?

A: We thank the reviewer for the comment.

To the best of our knowledge, studies on age and respiratory microbiome have focused on longitudinal studies of infants, and there are few references regarding direct comparisons of respiratory microbiome diversity between healthy young and old adults.

In a cross-sectional study of upper respiratory tract microbiota in elderly pneumonia patients, Piters et al. collected healthy elderly and young healthy adults as controls (**ISME J. 2016 Jan;10(1):97-108**). They determined the overall bacterial density of the samples, which was significantly higher in healthy elderly as opposed to healthy adults. But they lacked a comparison of the microbiota diversity between healthy elderly and healthy adults. We calculated the Shannon diversity on the raw data of this study, and the diversity was similar between healthy elderly and healthy adults.

Whelan et al. explained the loss of upper respiratory tract microbiota topography with age mainly due to population in the anterior nares of the elderly that more closely resembled the oropharynx (**Ann Am Thorac Soc. 2014 May;11(4):513-21**). Differences in diversity between the elderly and adults in the oropharynx were not indicated. And the elderly oropharyngeal microbiota were characterized by increased abundance and species of streptococci. This is consistent with our report.

Because the presence of more species of streptococci may be one of the reasons for the high

microbial diversity in the elderly. Microbial communities with higher diversity tend to have greater colonization resistance and stability, while abnormal propagation of pathogens in disease states can lead to decreased diversity. We believe that there is considerable stability of the oropharyngeal flora in the healthy elderly. This pattern has been validated in gut microbiome by numbers of studies. The higher gut microbial diversity in long-living people from two independent studies revealed a healthier gut microbiota in this group of people (**Figure 1B and C in Curr Biol. 2016 Sep 26;26(18):R832-R833**).

Specific comment:

1) In Result section (Line 218), it is mentioned that two cohort were established, including "age cohort" and "disease cohort". Is there longitudinal follow-up in the study? If not, please change it in the whole text.

A: Thanks.

"age cohort" and "disease cohort" had been changed to "age team" and "disease team" in the revised edition (**the first 2 sections of the result and Table 1**)

2) When comparing the association between heart failure and microbiome in the elderly, the author made age matching in the control group, but the gender difference was large (Table1). If the conclusion is established, please supplement the evidence supporting that gender has no effect on the microbiome in the elderly.

A: We thank the Reviewer for raising this point.

As shown in the following figure, we analyzed the microbial community structure of H group with large gender difference by PCA method, and found that most samples of F (female) group and M (male) group were clustered together. Previously, Whelan et al. also found that the microbiota of the elderly showed no association with sex (**Ann Am Thorac Soc. 2014 May;11(4):513-21**).

Re: Spectrum00216-24R1 (Oropharyngeal microbiome profiling and its association with age and heart failure in the elderly population from the northernmost province of China)

Dear Prof. Hong Ling:

Your manuscript has been accepted, and I am forwarding it to the ASM production staff for publication. Your paper will first be checked to make sure all elements meet the technical requirements. ASM staff will contact you if anything needs to be revised before copyediting and production can begin. Otherwise, you will be notified when your proofs are ready to be viewed.

Sincerely,
Zhe LYU
Editor
Microbiology Spectrum